# Structural insight into Marburg virus nucleoprotein–RNA complex formation

Yoko Fujita-Fujiharu [1,2,3], Yukihiko Sugita [1,2,4], Yuki Takamatsu[1,9], Kazuya Houri[1,2,3], Manabu Igarashi[5], Yukiko Muramoto [1,2,3], Masahiro Nakano [1,2,3], Yugo Tsunoda[1,2,3], Ichiro Taniguchi [6], Stephan Becker[7,8] & Takeshi Noda [1,2,3 ✉]

The nucleoprotein (NP) of Marburg virus (MARV), a close relative of Ebola virus (EBOV), encapsidates the single-stranded, negative-sense viral genomic RNA (vRNA) to form the helical NP–RNA complex. The NP–RNA complex constitutes the core structure for the assembly of the nucleocapsid that is responsible for viral RNA synthesis. Although appropriate interactions among NPs and RNA are required for the formation of nucleocapsid, the structural basis of the helical assembly remains largely elusive. Here, we show the structure of the MARV NP–RNA complex determined using cryo-electron microscopy at a resolution of 3.1 Å. The structures of the asymmetric unit, a complex of an NP and six RNA nucleotides, was very similar to that of EBOV, suggesting that both viruses share common mechanisms for the nucleocapsid formation. Structure-based mutational analysis of both MARV and EBOV NPs identified key residues for helical assembly and subsequent viral RNA synthesis. Importantly, most of the residues identified were conserved in both viruses. These findings provide a structural basis for understanding the nucleocapsid formation and contribute to the development of novel antivirals against MARV and EBOV.

[1] Laboratory of Ultrastructural Virology, Institute for Frontier Life and Medical Sciences, Kyoto University, 53 Shogoin Kawahara-cho, Sakyo-ku, Kyoto 606-8507, Japan. [2] Laboratory of Ultrastructural Virology, Graduate School of Biostudies, Kyoto University, 53 Shogoin Kawahara-cho, Sakyo-ku, Kyoto 606-8507, Japan. [3] CREST, Japan Science and Technology Agency, 4-1-8 Honcho, Kawaguchi, Saitama 332-0012, Japan. [4] Hakubi Center for Advanced Research, Kyoto University, Kyoto 606-8501, Japan. [5] Division of Global Epidemiology, International Institute for Zoonosis Control, Hokkaido University, Sapporo 001-0020, Japan. [6] Laboratory of RNA system, Institute for Frontier Life and Medical Sciences, Kyoto University, 53 Shogoin Kawahara-cho, Sakyo-ku, Kyoto 606-8507, Japan. [7] Institute of Virology, University of Marburg, 35043 Marburg, Germany. [8] German Center for Infection Research (DZIF), Marburg-Gießen-Langen Site, University of Marburg, 35043 Marburg, Germany. [9] Present address: Department of Virology I, National Institute of Infectious Diseases, Gakuen 4-7-1, Musashimurayama-city, Tokyo 208-0011, Japan. ✉email: t-noda@infront.kyoto-u.ac.jp

Marburg virus (MARV), a close relative of the Ebola virus (EBOV), belongs to the family *Filoviridae* and possesses a 19.1 kb non-segmented, single-stranded, negative-sense RNA genome (vRNA)[1]. Similar to EBOV, it causes severe hemorrhagic fever in humans and non-human primates with a high mortality rate, which continuously poses a threat to public health. However, no vaccine or antivirals against MARV diseases have yet been licensed. The nucleoprotein (NP) of MARV consisting of 695 amino acids encapsidates the vRNA to form a left-handed helical complex[2]. The helical NP–RNA complex acts as a scaffold for the formation of a nucleocapsid, which is responsible for the transcription and replication of the vRNA[3,4]. Since the NP–RNA complex is the structural unit that constitutes the helical core structure of the nucleocapsid, for which appropriate interactions between NP molecules and RNA nucleotides are required, solving the NP–RNA unit structure is essential for understanding the mechanisms of the nucleocapsid formation.

The core structure of RNA-free monomeric MARV NP, spanning residues 21–373, features a typical bilobed fold consisting of N-terminal and C-terminal lobes, as determined using X-ray crystallography at 2.9 Å resolution[5,6], which is similar to that of the EBOV NP core structure[7,8]. These two lobes are connected by a flexible hinge region and are considered to clamp the RNA strand between the two lobes by electrostatic interactions. The overall three-dimensional architecture of the NP–RNA complex, in which the cellular RNA was coated with truncated NP comprising its N-terminal 390 residues, was visualized using cryo-electron tomography (cryo-ET), showing that the helical NP–RNA complex constitutes the innermost layer of the MARV nucleocapsid[9]. However, since the resolution was limited to the scale of a few nanometers, the structural basis for the assembly of NP and RNA into the nucleocapsid core largely remains unknown.

Using single-particle cryo-electron microscopy (cryo-EM), we recently determined the structure of EBOV NP–RNA complex at 3.6 Å resolution[10]. Here, we employed the same technique to determine the cryo-EM structure of the MARV NP–RNA complex and identified key residues important for helical NP–RNA assembly and subsequent viral RNA synthesis from the nucleocapsid using structure-based mutagenesis.

## Results

### Overall structure of the MARV NP–RNA complex and its structural units.
Similar to EBOV NP, MARV NP possesses the structural domains: an N-terminal arm, an NP core composed of N- and C-terminal lobes, a disordered linker, and a C-terminal tail (Fig. 1a). To determine the structure of MARV NP associated with RNA, we expressed a C-terminally truncated NP which encompasses residues 1–395 containing the N-terminal arm and the NP core domains in mammalian cells to re-constitute the helical complex[9]. Purified complexes were imaged using cryo-EM (Supplementary Fig. 1a), and the structure was determined by single-particle analysis at 3.1 Å resolution (Fig. 1b, Supplementary Fig. 1b–d). The constituted complex shows a left-handed double-helical structure (Fig. 1b, Supplementary Fig. 1b), in which respective strands have similar dimensions: each strand contains single-strand RNA, and 30.50 NP subunits per turn and a 128.99 Å helical pitch (rise = 4.23 Å, rotation = 11.8052°) with the outer and inner diameters of ~330 Å and 255 Å, respectively. In the cryo-EM map, the density of the nucleotide bases and most of the side chains are clearly visible, which enabled us to build an atomic model unambiguously showing six RNA nucleotides enclosed by the N- and C-terminal lobes of NP (Fig. 1c–e). The respective asymmetric NP subunits, NP-a and NP-b, show very

similar conformations as shown by a 0.592 Å backbone Root Mean Square Deviation (RMSD) (Fig. 1c, Supplementary Fig. 1e). The models of these asymmetric subunits also fit into the NP region within a low-resolution cryo-ET reconstruction of the MARV nucleocapsid[11] (Supplementary Fig. 1f). Helix α16 confines the RNA strand into a cleft between the two lobes (Fig. 1d). The N-terminal arm (residues 1 to 19) protrudes sideway from the N-terminal lobe and has a short $3_{10}$-helix η1 (Fig. 1d), which is associated with a neighboring NP molecule within the helical complex. Overall, the MARV NP–RNA complex shares common structural features to that of the EBOV NP–RNA unit[10] (Supplementary Fig. 2), although the MARV NP–RNA complex reconstituted here is a double helix unlike the EBOV NP–RNA complex. Expression of the full-length wild-type MARV NP also showed the formation of a double-helical structure (Supplementary Fig. 3).

### Amino acid residues of MARV NP involved in interactions with RNA.
Comparison of our atomic model of the RNA-bound MARV NP with an RNA-free monomeric MARV NP structure[6] provides a possible explanation for the mechanism of RNA encapsidation (Fig. 2a). A structural alignment of our RNA-bound and the previously reported RNA-free NP[6] molecules exhibit a 0.885 Å backbone RMSD, suggesting no drastic conformational changes in the overall structure. Local conformational changes are observed mainly in the C-terminal lobe, probably to clamp the RNA strand. In particular, the short $3_{10}$-helix η6, which is disordered in the RNA-free state, appears underneath the RNA strand, possibly to make hydrogen bonds with the RNA (Fig. 2a, b). The C-terminal helix α15 is shifted outward of the helical complex so that the C-terminal helices α16 on the NP strands can be aligned like a zipper to capture the RNA in between the two lobes (Fig. 2a). Indeed, the electrostatic surface potential of the NP shows that the α15 and α16 helices confine the RNA in a positively charged cleft between the N- and C-terminal lobes (Supplementary Fig. 4).

In the positively charged cleft between the N- and C-terminal lobes, six RNA nucleotides are encapsidated with a "3-bases-inward, 3-bases-outward" configuration (Fig. 2c). Several polar residues are present within hydrogen-bonding distance of the RNA backbone and likely contact with the RNA bases within the cleft (Fig. 2c). In particular, the interactions between NP and RNA are predominantly electrostatic due to positively charged residues such as K142, K153, R156, and H292 with their side chains pointing to the phosphate backbone of the RNA (Fig. 2c, d). In contrast, the side chain of K230, which is reported not to be essential for the RNA binding[6], is oriented toward the RNA base (Fig. 2c, d). To further understand the mechanism of RNA binding, molecular dynamics (MD) simulation and binding free energy calculations were performed. The binding free energy of the six RNA bases (all of them modeled as uracil) within a single NP molecule (Supplementary Table 1) indicated that the total contribution of the RNA backbone in the NP-binding energy was larger than that of the RNA base, and the sixth uracil (U6) exhibited a relatively high binding energy, which was partially attributed to its interaction with the helix α7 of the adjacent NP. Also, the RNA backbone-specific interactions with the NP are consistent with a sequence-independent manner of the viral RNA encapsidation, similar to those of other negative-strand RNA viruses[10,12–19]. Multiple protein sequence alignment of filovirus NPs shows that most of the polar residues in the RNA-binding cleft are highly conserved among the family *Filoviridae* (Supplementary Fig. 5), suggesting that those residues play important roles in the RNA binding and/or the vRNA synthesis.

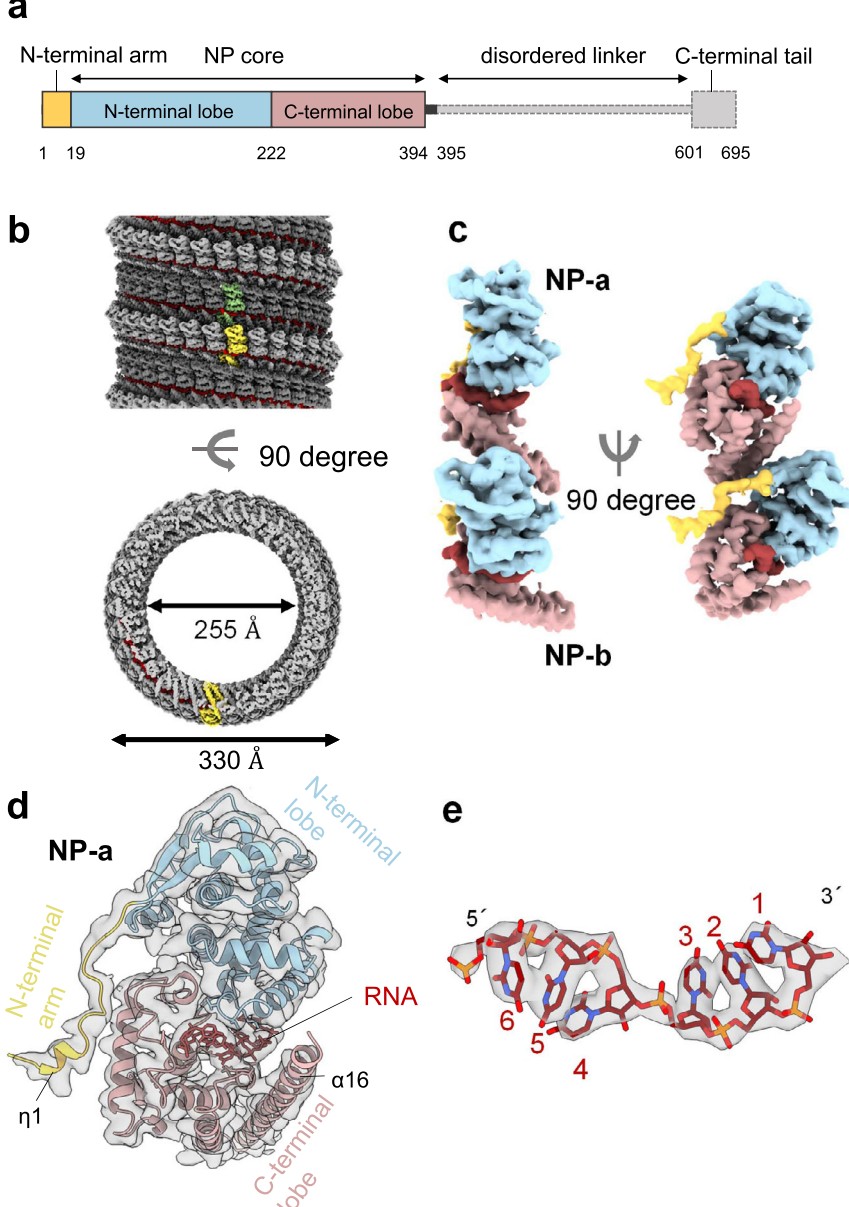

**Fig. 1 Overall cryo-EM structure and atomic model of MARV nucleoprotein (1–395). a** Structural layout of MARV NP sequence. The full-length NP consists of 695 amino acids and is divided into an N-terminal arm, an NP core consisting of N- and C-terminal lobes, a disordered region, and a C-terminal tail. **b** The iso-electron potential surface map of an NP–RNA complex reconstruction, calculated from 23,545 segments (contoured at 3σ above average). The RNA strand is highlighted in red. An NP-a molecule, which is in a dark gray strand, is highlighted in light green. An NP-b molecule in a light gray strand is highlighted in yellow. **c** The isolated NP–RNA complex unit (3σ) and **d** our atomic model with the EM map (3σ) in a ribbon representation, colored the same as in Fig. 1a. **e** Isolated EM map (6σ) of RNA superimposed with the atomic model. RNA bases are modeled as uracil.

**Amino acid residues involved in MARV NP–NP interactions.** The N-terminal arm is known to be essential for NP oligomerization of filoviruses[5–8,10,11,20]. However, the structural basis for the MARV NP–NP interactions remains elusive. Our atomic model shows that the N-terminal arm of any arbitrary NP molecule ($NP_n$) is associated with an adjacent $NP_{n+1}$ inside the helical complex (Fig. 3a). Similar to EBOV NP, the N-terminal arm of MARV $NP_n$ occupies a hydrophobic pocket located on the C-terminal lobe of $NP_{n+1}$ via L6 and L9 (Fig. 3b). Because the hydrophobic pocket of MARV NP is reportedly bound by N-terminal region of VP35, which blocks the NP oligomerization and RNA encapsidation[5,6], the interaction between the MARV N-terminal arm and the hydrophobic pocket is considered to be essential for the formation of the helical complex along with

RNA. Different from the structures of an RNA-free form of the NP bound to VP35 peptide[5,6], H4 in the N-terminal arm of $NP_n$ is tucked between K239 on the 240 loop and the hydrophobic pocket of $NP_{n+1}$ (Fig. 3b). A loop 202–208 on the N-terminal lobe of $NP_{n+1}$ slightly moves downward to make a new groove, which allows the N-terminal arm of $NP_n$ to enter this space (Fig. 3c). This local conformational change is probably caused by electrostatic interactions between D208 and D211 on $NP_{n+1}$ and R19 on the N-terminal arm of $NP_n$ (Fig. 3c).

In addition to the N-terminal arm and the N-terminal lobe, C-terminal helices of the NP are likely involved in interactions with a neighboring NP. In our model, helix α15 of $NP_n$ is associated with α16 of the adjacent $NP_{n+1}$, which is mediated by hydrophobic interactions between a hydrophobic patch

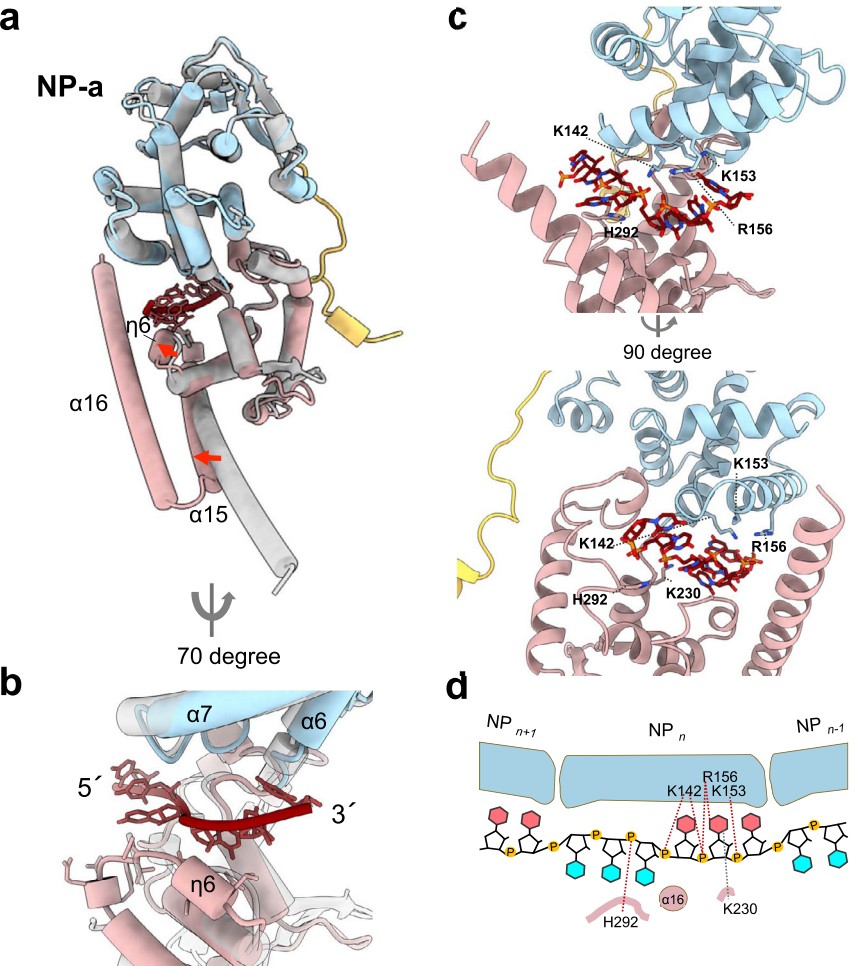

**Fig. 2 NP–RNA interactions. a** Overall structure of an RNA-bound NP molecule (PDB-ID: 7F1M, colored the same as Fig. 1a, from this study), which is superimposed with RNA-free monomeric NP (PDB-ID: 5F5M, gray). **b** Close-up view of the RNA binding site in a cylinder representation of α-helices and β-sheets. A secondary structure η6 appears underneath the RNA nucleotides. **c** Close-up view of the RNA binding site in a ribbon representation of α-helices and β-sheets, colored the same as Fig. 1a. **d** Schematic diagram of RNA recognition by positively charged residues of MARV NP. The RNA bases, which face inward and outward of the helix, are colored pink and cyan, respectively.

composed of L249, L325, and L336 on NP$_n$ and a hydrophobic region composed of I357, F361, and I368 on NP$_{n+1}$ (Fig. 3d). In addition, R339 on α15 of NP$_n$ points towards an acidic amino acid-rich region on the α16 of adjacent NP$_{n+1}$, suggesting that electrostatic interactions also occur between the α15 and α16 helices. This interaction might be more important for NP oligomerization than just for stably clamping the RNA strand within the helical complex as suggested previously[6]. Importantly, the residues described above, H4, L6, L9, R19, R339, and hydrophobic residues on helices α15 and α16, are highly conserved among the family *Filoviridae* (Supplementary Fig. 5), suggesting that interactions via the N-terminal arm and the C-terminal helices are important for NP oligomerization of filoviruses.

**Identification of amino acid residues important for filovirus helical assembly and subsequent viral RNA synthesis.** Having identified the potential interactions among MARV NPs and RNA as described above, we performed structure-based mutational analysis on MARV NP and examined the impact on the helical NP–RNA complex formation by negative staining EM. In parallel, we also assessed the corresponding residues of EBOV NP to understand the structural conservation among filoviruses. The

NP mutants tested are listed in Supplementary Table 2. Western blot analysis showed similar expression levels of all MARV and EBOV NP mutants in cells (Supplementary Fig. 7). Among the residues potentially involved in MARV NP–NP interactions, the H4A, L9E, R339A mutants formed similar helical NP–RNA complexes to wild-type MARV NP (1-395) (Fig. 4a). Similarly, the corresponding EBOV NP mutants, H22A, A27E, and Y357A formed helical NP–RNA complexes, which were morphologically indistinguishable from that of wild-type EBOV NP (1–450) (Fig. 4a). The MARV L6E mutant rarely showed helical complex formation, and when it formed, the helical complexes were significantly shorter than those of the wild-type (Supplementary Fig. 6), while the corresponding EBOV I24E mutant formed considerably loose helices, suggesting the involvement of the interaction between N-terminal arm and the hydrophobic pocket for appropriate NP oligomerization. Both MARV R19A mutant and its corresponding EBOV mutant, R37A, formed straighter and longer helical complexes, suggesting that interactions between the N-terminal arm and helix α10 affect rigidity of the helical complex (Supplementary Fig. 6). Among the residues potentially responsible for RNA binding, MARV R156A, K230A, and H292A, and the corresponding residues of EBOV NP R174A, K248A, and H310A formed helical NP–RNA complexes. In contrast, MARV K142A and K153A as well as corresponding

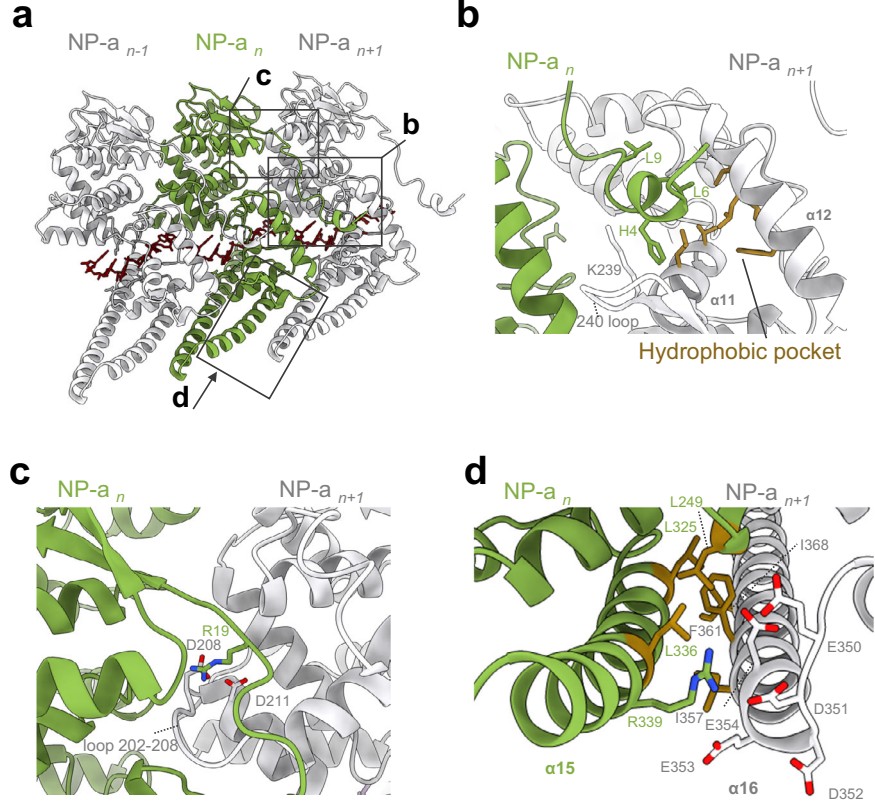

**Fig. 3 NP–NP interactions. a** Three adjacent NP molecules viewed from inside the helix. Black boxes show respective close-up figures. **b** NP–NP interactions between the N-terminal arm of NP$_n$ (green) and the hydrophobic pocket of an adjacent NP$_{n+1}$ (white, hydrophobic residues are shown in orange). **c** NP–NP interaction between the N-terminal lobe of NP$_n$ (green) and an adjacent N-terminal lobe of NP$_{n+1}$ (white). Basic residues are colored in blue, and acidic residues are colored in red. **d** NP–NP interaction between two C-terminal lobe helices in NP-a. Hydrophobic residues are shown in orange. Basic residues are colored in blue, and acidic residues are colored in red.

K160A and K171A in EBOV NP were not able to form regular helical complexes, suggesting their importance for the RNA binding property and for appropriate NP oligomerization.

Finally, to determine whether the amino acid residues that are responsible for appropriate helical NP–RNA complex formation are required for exerting nucleocapsid functions, we conducted minigenome assay using full-length MARV and EBOV NP mutants and assessed the impact of respective mutations on the transcription and replication activity. All of the NP mutants, which showed no morphological changes in the helical complexes (Fig. 4a), showed comparable levels of the transcription and replication activity to wild-type NP (Fig. 4b). In contrast, MARV L6E and R19A and the corresponding I24E, R37A of EBOV NP, which are involved in lateral NP–NP interaction, showed a significant reduction in their activity. In addition, MARV NP K142A and K153A and the corresponding K160A and K171A of EBOV NP, which are potentially involved in RNA binding, lost their activity to less than 1% of the wild-type. These results demonstrated that the amino acid residues, which are involved in appropriate helical NP–RNA complex formation, are essential for functional nucleocapsid formation.

## Discussion

MARV NP is an RNA-binding protein and a major component of the nucleocapsid. Here, we determined the structure of MARV NP–RNA complex at 3.1 Å resolution using single-particle cryo-EM. Despite substantial difference in the amino acid sequence, MARV NP shared common structural features with EBOV NP.

Each MARV NP encapsidated six nucleotides in between the N-and C-terminal lobes in a sequence-independent manner. The N-terminal arm is responsible for interaction with an adjacent NP. We also identified that amino acid residues of MARV and EBOV NPs critical for functional nucleocapsid formation as well as helical NP–RNA complex formation were mostly conserved in filoviruses.

Different from EBOV NP (1–450)-RNA complex, the MARV NP (1–395)-RNA complex purified from NP-expressing cells reproducibly constituted double-helical structures (Supplementary Fig. 8), although we used the same purification procedures for both[10]. Since the MARV nucleocapsid and its core structure is thought to be a single helix as shown by cryo-ET[9,11], the double-helical structure reconstituted in this study may not faithfully recapitulate the structure in the MARV nucleocapsid core. Nevertheless, the asymmetric subunits, NP-a and NP-b monomers, showed the same domain organization and structural features with NP molecules on the cryo-ET reconstruction of MARV[11] (Supplementary Fig. 1f). Furthermore, introduction of mutations into potential interaction regions among MARV NPs and RNA revealed changes in the NP-RNA helical formation and the nucleocapsid function (Fig. 4). These results indicate that not only the atomic coordinates of NP and RNA but also the interaction modes between lateral NPs in the helix and between NP and RNA nucleotides determined in this study reflect those in authentic MARV and in the virus-infected cells. Our structure also reveals flexible and yet stable structural features of the MARV NP to potentially form the metastable assemblies with RNA.

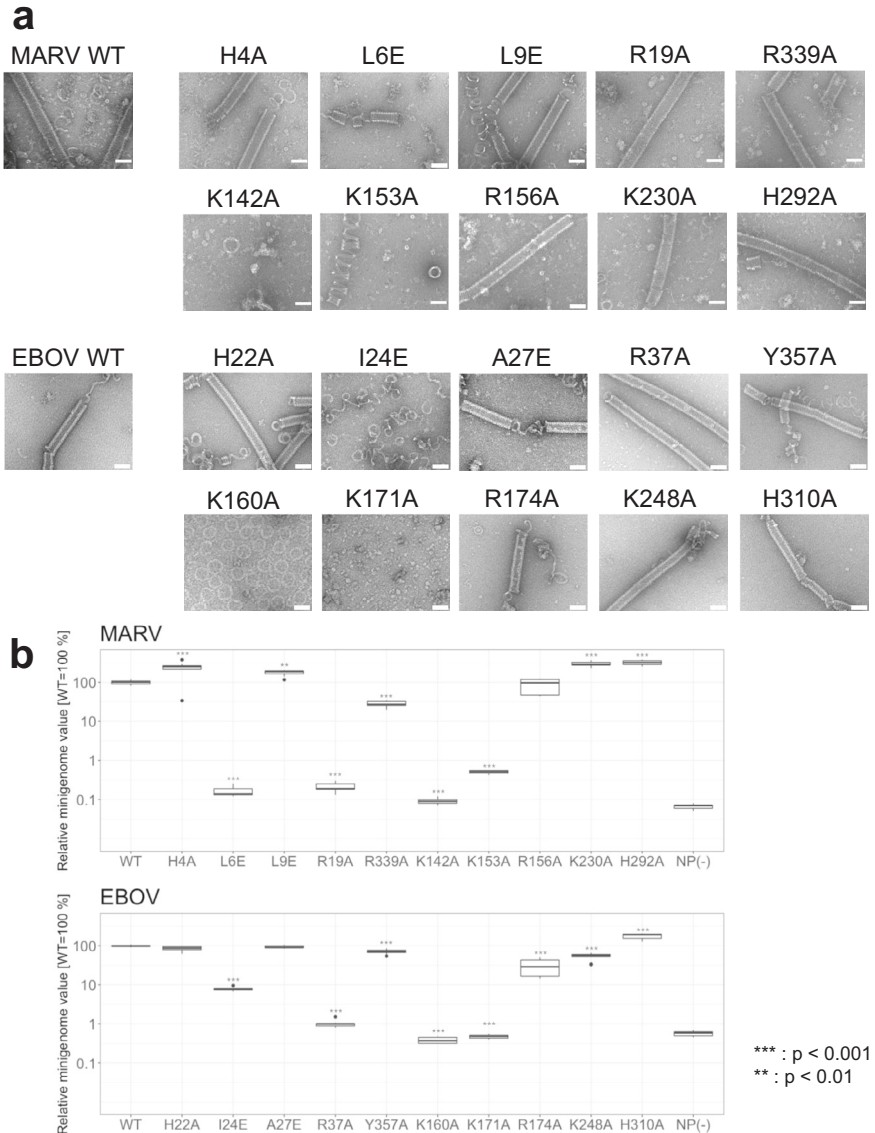

**Fig. 4 Site-directed mutagenesis analysis. a** A gallery of purified, negatively stained, helical complexes composed of C-terminally truncated MARV and EBOV NP mutant–RNA complexes and the wild-type (WT) complex. Scale bars, 50 nm. **b** Transcription and replication activities of MARV and EBOV NP mutants and the wild-type (WT), evaluated by minigenome assay. The experiments were performed in triplicates ($n = 3$). The statistical significance was tested by two-sided ANOVA-Dunnett's test to correct for multiple hypothesis testing. Box-and-whiskers plots represented the maxima, 75th percentile, median, 25th percentile, and minima, with a dot indicating the missing value. Source data are provided as a Source Data file.

It is believed that, in many RNA viruses, viral nucleoproteins bind to viral genomic RNA in a sequence-independent manner[10,12–19] which was also observed in our study (side chains of positively charged residues, K142, K153, R156, and H292, in the RNA-binding cleft pointed to the phosphate backbone of the RNA, but not to the RNA bases). The MD simulation followed by binding free energy analysis revealed that the energy contribution of RNA backbone to NP-binding was higher than that of the RNA bases. These results highlighted the importance of the interactions between RNA backbone and NP for RNA binding, which probably assists in the incorporation of viral RNA in a sequence-independent manner to form the helical NP–RNA complex.

Structure-based mutational analysis identified amino acid residues which are critical for helical complex formation and nucleocapsid function of EBOV and MARV. Regarding the RNA recognition, MARV K142 and K153 (corresponding to EBOV

K160 and K171, respectively) likely play more important roles in functional nucleocapsid formation, compared to K230 and H292 in the C-terminal lobe. Interestingly, paramyxovirus and pneumovirus nucleoproteins, which encapsidate vRNA in the positively charged cleft composed between the N- and C-terminal lobes, contain more basic amino acid residues on the RNA-binding cleft of N-terminal lobe than that of C-terminal lobe[12,15,21,22] (Supplementary Fig. 9). These findings indicate that robust association via N-terminal lobe and subsequent interaction of C-terminal lobe with RNA nucleotides would be important for nucleocapsid assembly and vRNA transcription and replication.

In our MARV NP–RNA complex structure, the N-terminal arm fits into the hydrophobic pocket of the adjacent NP (Fig. 3b, Supplementary Fig. 10), where the hydrophobic interaction plays a key role in tethering adjacent two NP molecules. Importantly, the amino acids comprising this hydrophobic pocket, such as

F223, V229, L233, L237, V244, L266, L269, A270, G273, A276, P277, and F278, are conserved among the family *Filoviridae* (Supplementary Fig. 5). The hydrophobic pocket is also occupied by the N-terminal region of MARV VP35[5,6] (Supplementary Fig. 10). Occupation of the hydrophobic pocket by the VP35 N-terminus prevents interaction between two adjacent NPs via the N-terminal arm, consequently hindering oligomerization. Thus, MARV NP would oligomerize by release of the VP35 N-terminus from the hydrophobic pocket and in turn bind to vRNA, as is the case with EBOV[7,8,10]. Although the N-terminal arm of MARV NP is 18-amino acids shorter than that of the Ebola and Cueva viruses (Supplementary Fig. 5), the competitive interactions with the hydrophobic pocket between the N-terminal arm and the VP35 N-terminus is probably a common NP oligomerization mechanism in the family *Filoviridae*[7,8,10].

In conclusion, we determined a high-resolution structure of MARV NP–RNA complex using cryo-EM and identified critical residues for functional nucleocapsid formation. The results advance our understanding of the mechanisms of the nucleocapsid formation and contribute to the development of antivirals broadly effective for filoviruses.

## Methods

**Cells**. Human embryonic kidney 293 Freestyle (HEK293F) cells were maintained in Expi293 Expression Medium (Gibco, Waltham, MA, USA), at 37 °C in an 8% CO₂ atmosphere. Human embryonic kidney (293T) cells were maintained in the Dulbecco's Modified Eagle Medium (D6046, Merck, Darmstadt, Germany) supplemented with 10% fetal bovine serum (FB-1365, Biosera, France).

**Plasmids**. All MARV (Musoke strain, GenBank ID YP_001531153.1) plasmids encoding wild-type proteins (pCAGGS-L, pCAGGS-VP30, pCAGGS-VP35, pCAGGS-NP), T7-driven MARV minigenome encoding *Renilla* luciferase (p3M-5M-Luc), and T7 DNA-dependent RNA polymerase (pCAGGS-T7), are described elsewhere[23]. The pCAGGS-NP (1–395) that expresses residues 1-395 of MARV NP was constructed from pCAGGS-NP by PCR using primers with digestion enzyme sites. Then, the PCR product was cloned into digested pCAGGS vector with a Kozak sequence present upstream of the initiation codon. The NP as well as NP (1–395) mutant constructs were generated by PCR-based site-directed mutagenesis. The primers used in this study are listed in Supplementary Table 3. All constructs were sequenced to confirm that unwanted mutations were not present.

**Expression and purification of the NP–RNA complex**. For MARV, HEK293F cells grown in 10 ml of medium (3.0 × 10⁶ cells/ml) were transfected with 6 µg of pCAGGS-NP (1–395) using Polyethyleneimine MAX (Polysciences, Warrington, PA, USA). Three days post-transfection, the cells were collected and lysed with 0.1% Nonidet P-40 substitute (Wako, Osaka, Japan) in Tris-HCl buffer (10 mM Tris-HCl (pH 8.0), 150 mM NaCl, 1 mM EDTA) including a cOmplete Protease Inhibitor (Roche, Basel, Switzerland) and 10 mM Ribonucleoside-Vanadyl Complex (NEB, Ipswich, MA, USA) on ice. The lysate was centrifuged at 10,000 *g* at 4 °C for 10 min to remove insoluble substances, and the supernatant was subjected to a discontinuous CsCl gradient ultracentrifugation at 246,100 *g* at 4 °C for 2 h. Then, fractions containing the NP–RNA complex were collected and ultracentrifuged at 246,100 *g* at 4 °C for 15 min. The pellet was suspended in Tris-HCl buffer.

For EBOV, HEK293T cells grown in 10 ml of medium (3.0 × 10⁵ cells/ml) were transfected with 10 µg of pCAGGS-NP (1–450) using *Trans*IT-293 Reagent (Takara, Shiga, Japan). Three days post-transfection, NP–RNA complexes were purified as described above and suspended in Tris-HCl buffer.

**Negative staining EM**. Five microliters of sample were applied to a glow-discharged copper grid coated with carbon and negatively stained with 2% uranyl acetate. Images were obtained with transmission electron microscopy (HT-7700 (system software v.02.22.15.15), Hitachi High Technology, Tokyo, Japan) operating at 80 kV with an XR81-B CCD camera. The length distribution was measured for about 60 helices by using ImageJ software[24,25].

**Cryo-EM specimen preparation**. The 2.5 µl sample of purified NP–RNA complex solution was applied onto glow-discharged 200-mesh R1.2/1.3 Cu grids (Quantifoil, Jena, Germany). The grids were blotted and rapidly frozen in liquid ethane using a Vitrobot Mark IV (Thermo Fisher Scientific, Waltham, MA, USA).

**Cryo-EM data collection**. Images were acquired on a Titan Krios cryo-EM (Thermo Fisher Scientific, Waltham, MA, USA) at 300 kV equipped with a Cs

corrector (CEOS, GmbH), which was installed in the Institute for Protein Research, Osaka University. Data were acquired automatically with EPU software as movies on a Falcon 3EC detector (Thermo Fisher Scientific, Waltham, MA, USA). The detailed imaging condition is described in Supplementary Table 4.

**Image processing**. We used the software RELION3.1[26] for the processing steps. The movie frames were motion-corrected using relion_motioncorrection with 5 × 5 patches. Next, GCTF[27] software was utilized for contrast transfer function (CTF) estimation of all micrographs. A total of 2469 micrographs were subjected to subsequent helical image processing. Helices were manually traced. A total of 30,668 segments were extracted and two rounds of 2D classification were performed. Classes, in which the detailed structure of helices was confirmed, were selected for further analysis. The 3D initial model, a featureless cylinder, was made by the relion_helix_toolbox command. Then, the featureless cylinder was refined via Refine3D and used as the initial model for the following 3D classification. The 3D classification was performed with local symmetry search around expected helical parameters from 2D class images (twist from 12.5° to 11°, rise from 4 Å to 4.5 Å). A total of 23,545 segments from the best resolved 3D class were subjected to 3D auto-refinement (helical parameter: twist for 11.8052°, rise for 4.23 Å). Finally, we ran the CTF refinement job and the Bayesian polishing job to improve the resolution and the final map was calculated from 3D auto-refinement. The resulting EM map has a nominal resolution of 3.1 Å estimated from the Fourier Shell Correlation and was normalized with PyMOL. Local resolution was estimated with RELION and the locally sharpened map was used for the following atomic model building. The detailed image processing conditions are listed in Supplementary Table 4.

**Atomic model building and refinement**. Atomic modeling of NP (1–395) was performed by template-based and de novo structure modeling. An atomic model of EBOV NP (1–450)–RNA complex (PDB-ID: 5Z9W)[10] served as the initial template. First, we performed a rigid body fitting of the atomic model of EBOV NP (1–450) into the locally sharpened EM map of MARV NP (1–395)-RNA complex using the software UCSF Chimera[28]. Next, the software COOT[29] was used to replace each amino acid of EBOV NPs with the amino acid sequence of MARV NPs; for amino acid residues from 389 to 391, which have no homologous sites in the atomic model of EBOV NPs, the models were constructed de novo. The atomic model was refined with phenix.real_space_refine[30]. To precisely refine the atomic models located in the intermolecular region, surrounding eight molecules of the atomic models were fitted to our map and refined with constraints on the protein secondary structure estimated by phenix_secondary_structure_restraints. The model validity was assessed by the software Phenix[30] and MolProbity[31]. The detailed results of the atomic model construction and its evaluation are shown in Supplementary Table 4. The software UCSF Chimera X[32] was used for displaying the EM map and the atomic models.

**Minigenome assay**. For MARV, each well of HEK 293T cells grown in a 12-well plate was transfected with the plasmids expressing MARV nucleocapsid components (400 ng of pCAGGS-L, 40 ng of pCAGGS-VP30, 40 ng of pCAGGS-VP35, 200 ng of pCAGGS-NP, and 400 ng of the minigenome p3M-5M-Luc), 200 ng of T7 polymerase (pCAGGS-T7), and 20 ng of firefly luciferase (pGL) by using 4 µl of TransIT 293 (Takara, Shiga, Japan) and 140 µl of OPTI-MEM (Thermo Fisher Scientific, Waltham, MA, USA). At 48 h post-transfection, the cells were lysed using lysis buffer (pjk, Kleinblittersdorf, Germany), and the *Renilla* and firefly luciferase activities were measured using GloMax®-Multi+ Detection System (Promega, Madison, WI, USA) with an Renilla-Juice kit and a FireFly-Juice kit (pjk, Kleinblittersdorf, Germany) according to the manufacturer's protocol. The *Renilla* luciferase activity was normalized to the firefly luciferase activity.

For EBOV, HEK293T cells were grown in a 24-well plate and each well was transfected with the plasmids expressing EBOV nucleocapsid components (1,000 ng of pCAGGS-L, 75 ng of pCAGGS-VP30, 100 ng of pCAGGS-VP35, 100 ng of pCAGGS-NP, and 200 ng of the minigenome p3E-5E-Luc), 200 ng of T7 polymerase (pCAGGS-T7), and 10 ng of Renilla luciferase (pTK r.luc) by using 4 µl of TransIT 293 (Takara, Shiga, Japan) and 100 µl of OPTI-MEM (Thermo Fisher Scientific, Waltham, MA, USA). At 48 h post-transfection, the cells were lysed using Passive Lysis buffer (Promega, Madison, WI, USA), and the firefly and Renilla luciferase activities were measured using GloMax®-Multi+ Detection System (Promega Madison, WI, USA) with a Luciferase Assay Reagent II and a Stop & Glo® Reagent (Promega Madison, WI, USA) according to the manufacturer's protocol. The firefly luciferase activity was normalized to the Renilla luciferase activity.

**Western blotting**. HEK293T cells, transfected with plasmids expressing MARV or EBOV nucleocapsid components, were dissolved with 2× Tris-Glycine SDS Sample Buffer (Thermo Fisher Scientific, Waltham, MA, USA), boiled for 5 min, and subjected to SDS-PAGE. Proteins were electroblotted onto membranes, and labeled with antibodies against MARV NP (10,000-fold dilution; polyclonal; kindly provided by Prof. Ayato Takada, Hokkaido University), EBOV NP (10,000-fold dilution; polyclonal; 0301-012, IBT BIOSERVECES, Rockville, MD, USA), and β-actin (10,000-fold dilution; monoclonal; #ab8226; Abcam, Cambridge, UK).

**Molecular dynamics simulation and binding free energy analysis**. MD simulation was performed for three consecutive units of MARV NP–RNA complex (Fig. 3a), each consisting of an NP and six RNA nucleotides, using the Amber 18 package[33]. The AMBER ff14SB[34] and RNA.OL3[35,36] force fields were used for the simulation. The protonation states of the ionizable residues were assigned a pH of 7.0 using the PDB2PQR web server[37]. The system was solvated in a truncated octahedral box of TIP3P water molecules with a distance of at least 10 Å around the proteins, and was then energy-minimized in four steps: first, only hydrogen atoms; second, water and ions; third, the side chains; and finally, all atoms. After energy minimization, the system was gradually heated from 0 to 310 K over 300 ps with harmonic restraints (with a force constant of 1.0 kcal/mol Å$^2$). Two additional rounds of MD (50 ps each at 310 K) were performed with decreasing the restraint weight from 0.5 to 0.1 kcal/mol Å$^2$. Next, 500 ns of unrestrained production run was performed in the NPT ensemble at a pressure of 1.0 atm and a temperature of 300 K. All bond lengths including hydrogen atoms were constrained using the SHAKE algorithm[38], and a time step of 2 fs was used. A cut-off radius of 10 Å was set for the nonbonded interactions. Long-range electrostatic interactions were treated using the particle mesh Ewald method[39]. The stability of the trajectory was assessed by monitoring the RMSD of the backbone Cα atoms from the initial structure of MD simulation. After confirming that RMSD in the system reached equilibrium within 200 ns, the trajectory extracted from the last 300 ns (i.e., 200–500 ns) was used for subsequent per-residue energy analysis. Binding free energies were calculated using the script of the molecular mechanics-generalized Born surface area (MM-GBSA) method in AmberTools18. The conformational entropy was not considered due to the high computational cost and low prediction accuracy. The salt concentration was set to 0.20 M.

**Statistical analysis and reproducibility**. Statistically significant differences in minigenome assay were evaluated using ANOVA-Dunnett's test to correct for multiple hypothesis testing with RStudio software. Statistically significant differences in the length of MARV NP variants were evaluated by unpaired Student $t$ test using RStudio software. Data are presented as the mean ± s.d. A $p$ value < 0.01 was considered statistically significant. The data from minigenome assay and negative staining analysis of MARV and EBOV NP mutants were repeated in three independent experiments.

**Reporting summary**. Further information on research design is available in the Nature Research Reporting Summary linked to this article.

## Data availability

The data that support this study are available from the corresponding author upon reasonable request. The cryo-EM maps generated in this study have been deposited into the Electron Microscopy Data Bank under accession code EMD-31420. The atomic coordinates reported in this paper have been deposited in the Protein Data Bank (PDB) with accession code 7F1M. Raw movies have been deposited in the Electron Microscopy Public Image Archive with accession code EMPIAR-10733. Previously released structural data used in the course of this study: 5Z9W and 5F5M were obtained from the PDB. Source data are provided with this paper.

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

## Acknowledgements

We thank Yoshihiro Kawaoka (Institute of Medical Science, University of Tokyo) for providing plasmids expressing EBOV viral proteins and minigenome, and So Iwata, as well as, Norimichi Nomura (Graduate School of Medicine, Kyoto University) for the pET-28a plasmids, and Ayato Takada (International Institute for Zoonosis Control, Hokkaido University) for the anti-MARV NP antiserum. This work was supported by a Grant-in-Aid for JSPS Fellows (21J12207) (to Y.F.), a Research Grant from the Kazato Research Encouragement Prize, MEXT Grant-in-Aid for Leading Initiative for Excellent Young Researchers, JSPS KAKENHI Grant Number 21K07052 (to Y.S.), JSPS KAKENHI Grant Number 19K16666, an AMED Research Program for Infectious Diseases Research and Infrastructure (Interdisciplinary Cutting-edge Research, 20wm0325023j0001, 21wm0325023j0002) (to Y.T.), JSPS KAKENHI Grant Number 20H03140 (to M.I.), JSPS KAKENHI Grant Numbers 20H03494, 19K22529, the JSPS Core-to-Core Program A, the Advanced Research Networks, MEXT Grant-in-Aid for Scientific Research on Innovative Area (19H04831), an AMED Research Program on Emerging and Re-emerging Infectious Disease grants (19fk0108113, 20fk0108270h0001; 21wm0325023j0002), the JST Core Research for Evolutional Science and Technology (JPMJCR20HA), the Grant for the Joint Usage / Research Center on Tropical Disease, Institute of Tropical Medicine, and Nagasaki University, the Daiichi Sankyo Foundation of Life Science, the Uehara Memorial Foundation (to T.N.), the Grant for Joint Research Project of the Institute of Medical Science, University of Tokyo, the Joint Usage/Research Center program of Institute for Frontier Life and Medical Sciences Kyoto University, and the Takeda Science Foundation (to Y.S. and T.N.). This study was also supported by the Platform Project for Supporting Drug Discovery and Life Science Research (Basis for Supporting Innovative Drug Discovery and Life Science Research (BINDS)) from AMED under Grant Number JP19am0101072 (support number 1632).

## Author contributions

Y.F., Y.S., and T.N. designed the study; Y.F., Y.S., Y.T., K.H., I.M., Y.M., M.N., Y.T, and I.T. performed experiments; Y.F., Y.S., S. B., and T.N. wrote the manuscript with input from all co-authors. All authors read and approved the final manuscript.

## Competing interests

The authors declare no competing interests.
