## [Peer Review File · Nature Communications]

Structural insight into Marburg virus nucleoprotein-RNA complex formationReviewers' Comments:

Reviewer #1:

Remarks to the Author:

The authors reported the cryo-EM structure of MARV NP core in complex with a single strand RNA. This structure explains the basic assembly of MARV NP nucleocapsid. Mostly, the MARV nucleocapsid is very similar to that of EBOV previously published. However, it is important to document the details of MARV NP nucleocapsid because it is an important human pathogen.

The main shortcoming in this report is the mutagenesis studies on NP and their interpretation.

1. The authors must show that K142A and K153A mutations did not reduce significantly the recombinant expression of MARV NP, and similarly for EBOV NP. In fact, if available, recombinant protein levels should be shown for all mutant NP.

2. If authors claim that mutant NP has reduced RNA binding, an RNA binding by monomeric mutant NP must be presented. The authors only presented data on the assembly of the NP-RNA complex. As shown in the same study, R156A and K230A mutations, which could reduce RNA binding, do not interfere with the assembly.

3. There are three aspects of NP function here; RNA binding, nucleocapsid assembly, and viral RNA synthesis. The experiments showed how the assembly was affected by these mutations. Without an appropriate assembly of the nucleocapsid, viral RNA synthesis was reduced because the nucleocapsid serves as the template. The authors should not present, as they did in the manuscript, that their mutations of NP reduce viral RNA synthesis. The correct interpretation should have been that their mutations of NP affected proper assembly of the nucleocapsid, which in turn reduced viral RNA synthesis because of lack of a functional template (i.e. functional nucleocapsid). This distinction is important because there are NP mutations which do not affect the nucleocapsid assembly, but reduce viral RNA synthesis. Authors did not introduce that type of NP mutations.

The missing data must be provided, and the interpretation of mutation result must be corrected before publication.

Minor comments:

"28 The NP-RNA complex serves as a scaffold for the assembly of the

29 nucleocapsid" What does this mean? The NP-RNA complex is the nucleocapsid.

"251 molecule switches to an "open-state" structure" Authors showed in this manuscript that MARV NP does not change from an "open-state" to a "closed-state". Having this phrase contradicts their own observation. The true mechanism is that occupation of the hydrophobic pocket by VP35 prevented interactions with the N-arm, and therefore prevented formation of the nucleocapsid.

"253 and NP binding to vRNA in turn," The grammar is not correct.

"254 of MARV is" "254 of MARV NP is"

Reviewer #2:

Remarks to the Author:

During the past decade there has seen a considerable volume of structural studies on the nucleocapsid and the ribonucleoprotein (RNP) complexes of filoviruses, with most findings coming from research conducted by using the Ebola virus (EBOV) nucleoprotein as model system. Notably, the work by Fugjita-Fujiharu and colleagues extends this body of knowledge by providing the first high-resolution structure of the Marburg virus (MARV) RNP, solved by means of cryo-EM single particle analysis and helical reconstruction. The described structure differs only in minor details from those available for EBOV, (as it would be expected from the close similarity between the respective amino acid sequences). Nonetheless, the authors performed a commendable work in pointing out and discussing and putting into context key similarities and subtle differences when occur. Overall, the paper is very

well written and illustrated, concise and very clear. Especially given the importance of MARV as re-emerging highly pathogenic virus and the continuous need for the scientific community to identify novel antiviral targets amenable for drug development against this group of viruses, it is opinion of this reviewer that the paper would be certainly of great interest for the readership of Nature Communications.

Minor flaws in the manuscript, as it appears in the present form, potentially derive from typos and little inconsistencies in the use of acronyms or metric units throughout the text, which have been highlighted by this reviewer as points for potential improvement of the article's - already high - quality (please, see below).

Suggested minor revisions before consideration for acceptance:

- Line 56: please, for completeness, add reference Zhu et al., 2017 J Virol. (doi:10.1128/JVI.00996-1) which has been cited by authors as reference n. 19
- Line 57: please, for completeness, add reference Leung et al., 2015 Cell Rep. (doi: 10.1016/j.celrep.2015.03.034.)
- Line 77: "to re-constitute"
- Fig.1a: please, substitute "A disordered linker" with "disordered linker"
- Line 83: 11.8052, please add the " ° " apex for angle degrees (as correctly done by authors in Table 4).
- Line 88: 0.592 Å (without dash) and better to write first the explication, then the acronym in bracket (RMSD)
- Line 101: would better sound as "provides a possible explanation for mechanism of RNA encapsidation"
- Line 103: 0.855 Å (without dash)
- Line 105-107: [...] lobe, probably to [...] the short [...]
- Line 110: [...] in a positively charged cleft
- Extended data, Fig.5: please, consider using the EBOV acronym instead of the old ZEBOV in the alignment figure
- Line 153: L336 instead of "I336"
- Line 271: I guess that authors mean "Musoke strain" instead of "Musoki"; please, also add GenBank or NCBI protein or nucleotide database code for that
- Line 287: "Basel"

Minor suggestion/comment #1: if possible, this reviewer would be pleased to see the Ravn virus NP sequence included in the Extended data alignment of Fig.5. Although fully conserved in the residue position that have been highlighted here as important for NP-RNA binding and NP-NP interactions, yet the two sequences should differ at positions 94, 102 and 245 of the NPcore.

Minor suggestion/comment #2: unless this is a stylistic choice of the authors, there should not be any need to explicate acronyms in every figure title and/or caption. If they want, authors are encouraged to directly use acronyms there (e.g. MARV, EBOV, cryo-EM, NP-RNA etc.)

REVIEWER COMMENTS

Reviewer #1 (Remarks to the Author):

The authors reported the cryo-EM structure of MARV NP core in complex with a single strand RNA. This structure explains the basic assembly of MARV NP nucleocapsid. Mostly, the MARV nucleocapsid is very similar to that of EBOV previously published. However, it is important to document the details of MARV NP nucleocapsid because it is an important human pathogen. The main shortcoming in this report is the mutagenesis studies on NP and their interpretation.

Response: We appreciate constructive comments and suggestions from the reviewer #1. In response to the reviewer's comments, we performed several experiments and changed our interpretation as described below.

1. The authors must show that K142A and K153A mutations did not reduce significantly the recombinant expression of MARV NP, and similarly for EBOV NP. In fact, if available, recombinant protein levels should be shown for all mutant NP.

Response: As suggested by the reviewer, we performed western blotting using all MARV and EBOV mutants and confirmed that their expression levels are comparable to wild type NPs. We added the description in lines 169-170 and the data in Extended Figure 7.

2. If authors claim that mutant NP has reduced RNA binding, an RNA binding by monomeric mutant NP must be presented. The authors only presented data on the assembly of the NP-RNA complex. As shown in the same study, R156A and K230A mutations, which could reduce RNA binding, do not interfere with the assembly.

Response: As suggested by the reviewer, we generated monomeric MARV NP mutants, which comprise amino acid residues 19 to 370 (Supporting reference 1), performed electrophoretic mobility shift assay (EMSA) using in vitro-synthesized RNAs, and evaluated their RNA binding property (Supporting Figure 1). WT NP and the H292A mutant, which form helical NP-RNA complexes, resulted in an apparent band shift and the R156A and K230A mutants, which also form helical NP-RNA complexes, produced smear bands with a slower migration. In contrast, the K142A and K153A mutants, which did not form

appropriate helical NP-RNA complexes, showed no band shift. Thus, our EMSA data were somewhat consistent with the results of helical NP-RNA assembly obtained by EM (Fig. 4a) and imply that K142 and K153 are important for RNA binding.

However, it remains unclear why the mutants showed two different slower migration patterns (tight bands for the WT and H292A mutant, and smear bands for the K156A and K230A mutants), despite that the NP mutants used here were monomeric, as confirmed by gel filtration chromatography and EM (Supporting Figure 2, 3). Accordingly, because we were not able to obtain clear results regarding the RNA binding property of each NP mutant, we prefer not to show the EMSA data in the manuscript.

Thus, in response to the reviewer's comment, we revised the wording in the mutagenesis section as follows:

Line 68-70, We reworded "important for NP-RNA and NP-NP interactions" to "important for helical NP-RNA assembly"

Line 163-164, We reworded "NP-RNA binding" to "helical assembly"

Line 197, We reworded "which are involved in RNA binding" to "which are potentially involved in RNA binding"

Supporting Figure 1 EMSA using MARV NP (19-370) and in vitro-synthesized RNA

Supporting Figure 2
Size exclusion chromatograms for MARV NP (19-370)

Supporting Figure 3 Negative Staining of purified MARV NP (19-370)

Supporting Methods:

1. Expression and purification of NP (19-370)

pET-28a-MARV NP (19-370) was transformed into E. coli strain Rosetta TM(DE3)pLysS Competent Cells (Merck, Darmstadt, Germany). Transformed cells were cultured at 37 °C to OD₆₀₀ ~0.6 and were transferred to 16 °C. Then, protein was induced with 0.25 mM IPTG for an additional 16 h in culture. Cell pellets were suspended in 20 mM HEPES (pH 7.5), 500 mM NaCl, and 5% (vol/vol) glycerol buffer, and lysed by sonication and clarified by centrifugation at 25,000 × g for 30 min at 4 °C. The supernatant was subjected to Ni-NTA Superflow (Qiagen, Hilden, Germany) performed by gravity flow, washed with 20 mM HEPES (pH 7.5), 500 mM NaCl, 5% (vol/vol) glycerol, and 25 mM imidazole buffer, and then eluted with 20 mM HEPES (pH 7.5), 500 mM NaCl, 5% (vol/vol) glycerol, and 500 mM imidazole buffer. The protein was further purified by Superdex 200 increase 10/300GL size-exclusion column (Cytiva, Tokyo, Japan) pre-equilibrated with the gel filtration buffer (20 mM Tris-HCl (pH 8.0), 200 mM NaCl, 10% (vol/vol) glycerol, 1 mM DTT, and 0.1 mM EDTA).

2. Preparation of ³²P labelled RNA substrates

Templates for ³²P labelled RNA substrates (eGFP 1-150 nt) were amplified by PCR using

primers with a T7 phage promoter sequence (TAATACGACTCACTATAGGG) and transcribed with ³²P-UTPs according to manufacturer instructions. After 1 h of incubation at 37 ° C the mixture was extracted with phenol/chloroform and unincorporated NTPs were eliminated by a Sephadex G-50 spin column (Merck, Darmstadt, Germany).

3. Electrophoretic mobility shift assays of NP (19-370)

³²P-labeled RNAs (20,000 cpm) were mixed with purified monomeric NP protein (at the final concentration of 7 μM) in the presence of yeast tRNA (1 mg/ml) and incubated at 20 ° C for 15 min. The mixtures were loaded onto a 6% native polyacrylamide gel in 0.5x TBE buffer for electrophoresis. The bands were visualized by autoradiography.

Supporting reference.

1. Liu, B., Dong, S., Li, G., Wang, W., Liu, X., Wang, Y., Yang, C., Rao, Z., & Guo, Y. (2017). Structural Insight into Nucleoprotein Conformation Change Chaperoned by VP35 Peptide in Marburg Virus. *Journal of virology*, 91(16), e00825-17

3. There are three aspects of NP function here; RNA binding, nucleocapsid assembly, and viral RNA synthesis. The experiments showed how the assembly was affected by these mutations. Without an appropriate assembly of the nucleocapsid, viral RNA synthesis was reduced because the nucleocapsid serves as the template. The authors should not present, as they did in the manuscript, that their mutations of NP reduce viral RNA synthesis. The correct interpretation should have been that their mutations of NP affected proper assembly of the nucleocapsid, which in turn reduced viral RNA synthesis because of lack of a functional template (i.e. functional nucleocapsid). This distinction is important because there are NP mutations which do not affect the nucleocapsid assembly, but reduce viral RNA synthesis. Authors did not introduce that type of NP mutations.

The missing data must be provided, and the interpretation of mutation result must be corrected before publication.

Response: We agree with this reviewer's comment that the mutations, which inhibit appropriate helical assembly, cause a reduction in viral RNA synthesis due to the inability to form functional helical nucleocapsids. Because there were no mutations that did not affect the helical assembly, but reduced viral RNA synthesis in our analysis, we corrected the interpretation and revised the wording in the manuscript as follows:

Lines 37, We reworded the sentence to “Structure-based mutational analysis of both MARV and EBOV NPs identified key residues for helical assembly and subsequent viral RNA synthesis.”

Lines 68-70, We reworded the sentence to “we aimed to identify key residues important for helical NP-RNA assembly and subsequent viral RNA synthesis from the nucleocapsid”

Lines 163-164, We reworded the sentence to “Identification of amino acid residues important for filovirus helical assembly and subsequent viral RNA synthesis”

Minor comments:

Lines 28-29 “The NP-RNA complex serves as a scaffold for the assembly of the nucleocapsid” What does this mean? The NP-RNA complex is the nucleocapsid.

Response: As described in lines 46-49, we defined a nucleocapsid as a complex that is responsible for viral RNA synthesis. For clarity, we have revised the sentence to “The NP-RNA complex constitutes the core structure for the assembly of the nucleocapsid that is responsible for viral RNA synthesis” in lines 29-30.

Lines 251-253 “molecule switches to an “open-state” structure” Authors showed in this manuscript that MARV NP does not change from an “open-state” to a “closed-state” . Having this phrase contradicts their own observation. The true mechanism is that occupation of the hydrophobic pocket by VP35 prevented interactions with the N-arm, and therefore prevented formation of the nucleocapsid.

Response: The reviewer is correct. We have revised the sentence to “Occupation of the hydrophobic pocket by the VP35 N-terminus prevents interaction between two adjacent NPs via the N-terminal arm, consequently hindering oligomerization.” (lines 252-254)

Line 253 “and NP binding to vRNA in turn,” The grammar is not correct.

Response: We have corrected the grammatical error to “MARV NP would oligomerize by release of the VP35 N-terminus from the hydrophobic pocket and in turn bind to vRNA” . (lines 254-255)

Line 254 “of MARV is” is “of MARV NP is”

Response: We have corrected the typographical error (line 256).

Reviewer #2 (Remarks to the Author):

During the past decade there has seen a considerable volume of structural studies on the nucleocapsid and the ribonucleoprotein (RNP) complexes of filoviruses, with most findings coming from research conducted by using the Ebola virus (EBOV) nucleoprotein as model system. Notably, the work by Fugjita-Fujiharu and colleagues extends this body of knowledge by providing the first high-resolution structure of the Marburg virus (MARV) RNP, solved by means of cryo-EM single particle analysis and helical reconstruction. The described structure differs only in minor details from those available for EBOV, (as it would be expected from the close similarity between the respective amino acid sequences). Nonetheless, the authors performed a commendable work in pointing out and discussing and putting into context key similarities and subtle differences when occur. Overall, the paper is very well written and illustrated, concise and very clear. Especially given the importance of MARV as

re-emerging highly pathogenic virus and the continuous need for the scientific community to identify novel antiviral targets amenable for drug development against this group of viruses, it is opinion of this reviewer that the paper would be certainly of great interest for the readership of Nature Communications.

Minor flaws in the manuscript, as it appears in the present form, potentially derive from typos and little inconsistencies in the use of acronyms or metric units throughout the text, which have been highlighted by this reviewer as points for potential improvement of the article' s - already high - quality (please, see below).

Response: We are very grateful to Reviewer #2 for the careful reading and positive evaluation of our manuscript. As suggested by the reviewer, we have revised our manuscript and added extended data as described below.

Suggested minor revisions before consideration for acceptance:

- Line 56: please, for completeness, add reference Zhu et al., 2017 J Virol. (doi:10.1128/JVI.00996-1) which has been cited by authors as reference n. 19

Response: We have cited the report as reference #5 in the revised manuscript (line

57).

- Line 57: please, for completeness, add reference Leung et al., 2015 Cell Rep. (doi: 10.1016/j.celrep.2015.03.034.)

Response: We have added the report as reference #8 (line 58).

- Line 77: “to re-constitute”

Response: We have corrected the word to “re-constitute” (line 78).

- Fig. 1a: please, substitute “A disordered linker” with “disordered linker”

Response: We have corrected the term in Fig 1a.

- Line 83: 11.8052, please add the “ ° ” apex for angle degrees (as correctly done by authors in Table 4).

Response: We have added the “ ° ” in line 84.

- Line 88: 0.592 Å (without dash) and better to write first the explication, then the acronym in bracket (RMSD)

Response: We have revised to “0.592 Å backbone Root Mean Square Deviation (RMSD)” . (Line 88-89)

- Line 101: would better sound as “provides a possible explanation for mechanism of RNA encapsidation”

Response: As suggested by the reviewer, we have revised the sentence to “provides a possible explanation for the mechanism of RNA encapsidation” in lines 102-103.

- Line 103: 0.855 Å (without dash)

Response: We have deleted dash (line 104).

- Line 105-107: [...] lobe, probably to [...] the short [...]

Response: As suggested by the reviewer, we have revised the sentence to “Local conformational changes are observed mainly in the C-terminal lobe, probably to clamp the RNA strand. In particular, the short 3_{10} -helix $\eta 6$, which is disordered in the RNA-free state, appears underneath the RNA strand, possibly to make hydrogen bonds with the RNA” in lines 105-108.

- Line 110: [...] in a positively charged cleft

Response: We have revised to “in a positively charged cleft” in line 112.

- Extended data, Fig. 5: please, consider using the EBOV acronym instead of the old ZEBOV in the alignment figure

Response: We have corrected the term to EBOV in Extended Figure 5.

- Line 153: L336 instead of “I336”

Response: We have corrected the typographical error in line 153.

- Line 271: I guess that authors mean “Musoke strain” instead of “Musoki” ; please, also add GenBank or NCBI protein or nucleotide database code for that

Response: We have corrected the typographical error and added the GenBank ID in line 273.

- Line 287: “Basel”

Response: We have corrected the typographical error.

Minor suggestion/comment #1: if possible, this reviewer would be pleased to see the Ravn virus NP sequence included in the Extended data alignment of Fig. 5. Although fully conserved in the residue position that have been highlighted here as important for NP-RNA binding and NP-NP interactions, yet the two sequences should differ at positions 94, 102 and 245 of the NPcore.

Response: We appreciate the important comment by the reviewer. As suggested, we have added the Ravn virus NP sequence in Extended Figure 5.

Minor suggestion/comment #2: unless this is a stylistic choice of the authors, there should not be any need to explicate acronyms in every figure title and/or caption. If they want, authors are encouraged to directly use acronyms there (e. g. MARV, EBOV, cryo-EM, NP-RNA etc.)

Response: As suggested by the reviewer, we used acronyms in all figure legends.

Reviewers' Comments:

Reviewer #1:

Remarks to the Author:

The addition of supplement data and revision of the manuscript are satisfactory to me.